**Data Availability Statement:** Data in it's identified form cannot be shared publicly because of anonymity and privacy requirements. Data are available from the University of California, San

# "Chasing the pain relief, not the high": Experiences managing pain after opioid reductions among patients with HIV and a history of substance use

**Emily Behar**[1,2]*, **Rita Bagnulo**[1], **Kelly Knight**[2], **Glenn-Milo Santos**[1,2], **Phillip O. Coffin**[1,2]

**1** San Francisco Department of Public Health, San Francisco, California, United States of America,
**2** University of California, San Francisco, San Francisco, California, United States of America

* Emily.Behar@sfdph.org

## Abstract

### Background

Opioid overdose mortality continues to increase in the United States despite significant investments to reverse the epidemic. The national response to-date has focused primarily on reducing opioid prescribing, yet reductions in prescribing have been associated with patients reporting uncontrolled pain, psychological distress, and transition to illicit substances. The aim of this study is to qualitatively explore chronic pain management experiences among PLWH with a history of illicit substance use after long-term opioid therapy reductions or discontinuations.

### Methods

We analyzed 18 interviews, stopping upon reaching thematic saturation, with HIV-positive participants with a history of substance use who were enrolled in a longitudinal cohort study to assess the impact of prescribing changes among patients with chronic pain. Participants in this nested qualitative study had been reduced/discontinued from opioid pain relievers (OPRs) within the 12 months prior to interview. Interviews were audio-recorded and transcribed verbatim. Two analysts coded all interviews, interrater reliability was measured, and coding discrepancies discussed. The study took place in San Francisco, California in 2018.

### Results

Eleven participants were male with a mean age of 55; 8 were African American and 8 were White. All participants were HIV-positive, actively engaged in primary care, and had a lifetime history of illicit substance use. Twelve reported using illicit substances within the past year, including non-prescription opioids/heroin (10), and stimulant use (10). After being reduced/discontinued from their long-term opioid therapy, patients reported developing complex multimodal pain management systems that often included both nonpharmacological approaches and illicit substance use. Participants encountered a range of barriers to nonpharmacological therapies including issues related to accessibility and availability.

Francisco Institutional Data Access / Ethics Committee (contact via telephone at 415-476-1814, or email at compliance@ucsf.edu) for researchers who meet the criteria for access to confidential data.

**Funding:** This research was supported by a grant from the National Institutes of Health, University of California, San Francisco, Center for AIDS Prevention Studies, #P30MH062246 (ML, https://globalhealthsciences.ucsf.edu/international-traineeships-aids-prevention-studies-itaps) and a grant from the National Institute on Drug Abuse at the National Institutes of Health, #1R01DA040189 (PC, https://www.drugabuse.gov/). The funders had no role in study design, data collection and analysis, decision to publish, or preparation of the manuscript.

**Competing interests:** The authors have declared that no competing interests exist.

Participants often reported attempts to replicate their prior OPR prescription by seeking out the same medication and dose from illicit sources and reported transitioning to heroin after exhausting other options.

## Conclusion

After being reduced/discontinued from OPRs, HIV-positive patients with a history of substance use reported experimenting with a range of pain management modalities including nonpharmacological therapies and illicit substance use to manage symptoms of opioid withdrawal and pain. Providers should consider that any change to a patients' long-term opioid therapy may result in experimentation with pain management outside of the medical setting and may want to employ patient-centered, holistic approaches when managing patients' opioid prescriptions and chronic pain.

## Background

Opioid overdose mortality continues to increase in the United States despite significant investments to reverse the epidemic. The national response to-date has focused primarily on reducing prescribing of opioid pain relievers (OPRs), based on evidence that long-term OPR therapy has no greater effect on chronic non-cancer pain than acetaminophen or non-steroidal anti-inflammatory agents, and carries greater risks [1,2]. However, decreasing opioid dose or prescription cessation also carries risks: patients may experience uncontrolled pain, psychological distress, transitions to illicit substances, and even suicide [3–7]. Furthermore, due to increases in heroin and fentanyl availability, mortality has increased concordant with reductions in OPR prescribing [3,7,8].

The Centers for Disease Control and Prevention (CDC) Opioid Prescribing Guidelines recommend against using OPRs as first line therapy for managing chronic pain, and instead endorse nonpharmacological therapies such as behavioral, movement-based or integrative therapies [2]. While evidence about the effectiveness of these modalities is limited compared to opioid pharmacotherapy, a systematic review of noninvasive nonpharmacological treatments for chronic pain suggests that exercise, multidisciplinary rehabilitation, acupuncture, cognitive behavioral therapy, and mind-body practices were associated with improvements in pain and function among patients with selected chronic pain conditions [9]. Nonetheless, there remain a multitude of barriers to accessing these therapies, such as insufficient insurance coverage, limited referral options, and logistical challenges [10–13]. Moreover, the CDC guidelines are focused on opioid-naïve patients and authors recently clarified that the dose limitations and related recommendations were never intended to apply to patients already maintained on long-term OPRs [14]. Retrospective research has found that patients with a history of substance use [15–17], on a high daily dose of opioids [2], and with multiple pain complaints may be at heightened risk of transitioning to illicit opioids [18]; some of these transitions may be iatrogenically facilitated by tapering interventions or other measures to suppress access to pharmacotherapy [19].

People living with HIV (PLWH) may be at particularly high-risk in the context of changing prescribing practices, as they suffer from high rates of multiple medical disorders that increase the likelihood of chronic pain, the risks associated with OPR therapy, and social and environmental challenges [20]. PLWH also have unique causes of pain, such as HIV-associated neuropathy, and higher prevalence of substance use disorders and drug injection which may also contribute to higher chronic and acute pain [21]. Moreover, early HIV care was associated with palliative care medicine, thus many patients have already been provided OPRs long-term

[22,23]. To date, there is a dearth of qualitative research exploring how long-term opioid reductions affect this vulnerable population of PLWH with a history of substance use. The aim of this study is to qualitatively explore chronic pain management experiences among PLWH with a history of illicit substance use after long-term opioid therapy reductions or discontinuations to better understand patients' rationales for and descriptions of barriers and facilitators to pain management modalities.

## Methods

### Study sample

Participants were recruited in 2018 from a longitudinal cohort study, COPING (Cohort study of Opioids, Pain, and safety IN an era of chanGing policy), which aimed to assess the impact of prescribing changes in pain, functional status, prescribed opioid use, illicit substance use and opioid overdose risks among patients with chronic pain (N = 300) in San Francisco. Participants in COPING were recruited from safety-net clinics, English-speaking, ≥18 years old, had been prescribed long-term opioids (≥3 months) for chronic non-cancer pain for at least three of the 12 months prior to enrollment, and had a history of illicit opioid, cocaine, or methamphetamine use. Participants' opioid prescribing history was determined through an extensive medical record chart abstraction conducted by study staff. Participants for this nested qualitative study were also HIV-positive and had been reduced or discontinued from OPRs in preceding 12 months, as determined by a medical chart review.

### Data collection and analysis

We conducted 18 semi-structured interviews with ongoing interim analyses throughout the interview process to determine when theoretical saturation was reached [24,25]. An interviewer trained in qualitative methods (EB) conducted the interviews which lasted approximately 45 minutes and took place at the San Francisco Department of Public Health. Participants provided oral consent which was documented in a consent log. Participants were compensated $30. Study procedures were approved by the University of California, San Francisco, Institutional Review Board (IRB#15–18274).

Interviews were audio-recorded and transcribed verbatim, after which data were entered into Atlas.ti (Version 8). We used content analysis to expose emergent data through descriptive summaries [26,27]. This method of qualitative description, rooted in phenomenology, is best suited for moments of early exploration when outcomes are focused on exposition instead of hypothesis or theory generation [28,29]. Two analysts (EB, RB) independently reviewed the interviews and extracted emergent themes to inform the development of a master codebook. *A priori* and inductively-generated codes were compared and discrepancies discussed until consensus was formed and the codebook finalized. Analysts then coded the interviews and measured interrater reliability. Upon completion of the process, results were organized into thematic findings.

## Results

### Demographics

Eleven participants were male with a mean age of 55; 8 were African American and 8 were White. All participants were HIV-positive, actively engaged in primary care, and had a lifetime history of illicit substance use. Twelve participants reported using illicit substances within the past year, including illicit opioids (OPRs or heroin, 10), and stimulants (10) (Table 1).

**Table 1. Demographics and substance use (N = 18).**

|  | N | % |
|---|---|---|
| **Gender** | | |
| Male | 11 | 61% |
| Female | 6 | 33% |
| Transgender | 1 | 6% |
| **Race** | | |
| Black | 8 | 44% |
| White | 8 | 44% |
| Other | 2 | 11% |
| **Age, mean** | 55 | |
| **Substance Use, lifetime** | | |
| Any | 18 | 100% |
| Illicit OPRs/heroin | 11 | 61% |
| Stimulants | 14 | 78% |
| **Substance Use, last 12 months** | | |
| Any | 12 | 67% |
| Illicit OPRs/heroin | 10 | 56% |
| Stimulants | 10 | 56% |

Participant's opioid reduction and/or discontinuation was evaluated by chart abstraction and participant self-report but was only measured qualitatively; we did not differentiate between prescription reduction versus discontinued, nor did we measure the mean reduction amount. Overall, the vast majority of participants neither connected their pain to their HIV, nor reported that their HIV disease or care were affected by changes in OPR prescriptions. Because of this, even though the population was comprised of PLWH, our analysis did not result in HIV-specific themes. In the results below, we describe four strategies that participants reported using to manage their pain after being reduced/discontinued from opioids: (1) non-pharmacological therapies, (2) illicit OPRs, (3) heroin, and (4) stimulants.

## Nonpharmacological therapies

**Rationale for use.** Most participants reported utilizing a range of nonpharmacological therapies to manage their pain, including physical therapy, acupuncture, massage, yoga, prayer, reading, writing, and attending social support groups. Some participants reported using these therapies because of prior exposure in a clinical setting, as Participant A, a white transwoman in her early-50s, described:

Interviewer (I): Is there anything else you're doing for your pain?

Respondent (R): Just my physical and occupational therapy.

I: Okay. And how often do you do those?

R: They gave me. . .I don't have the therapist come visit anymore but I still do the exercises every day.

Other participants described using nonpharmacological therapies because they were legal. For instance, Participant B, a black man in his early-60s, reported refraining from illicit opioid use for fear of jeopardizing his active prescription:

I tried to do everything right [to manage pain]. I didn't go to the street to cop some, even though I wanted to. I wanted to go to the street so bad and cop me some, try to get me some pills. . .I said, "I'm not gonna do nothing wrong."

In lieu of illicit opioids, Participant B reported exercising, weight lifting, and attending physical therapy to manage his pain.

**Facilitators/Benefits.**   The majority of participants who used nonpharmacological therapies found that they reduced both pain and opioid intake. In addition, participants noted ancillary benefits, such as general enjoyment of the intervention and psychological improvements. Participant F, a black man in his mid-50s, explained his positive experience using exercise as a means of pain management: "Exercise and stretching and walking. I walk a lot. I do like to walk. . .it also helps this little body of mine keep moving. . .I love that part."

Participant G, a black woman in her late-60s, described the benefits of passive nonpharmacological approaches that she discovered on her own: "I read, yeah, I read, I listen to music, and sometimes I just walk. Whenever I'm in pain, I just get up and walk around the block, take my mind off of it or whatever." In addition, she reported benefiting from an HIV support group even though it was not focused on pain management:

I go to these support groups . . .And they're not really about pain, you know, but it . . . helps me. . .It don't need me to just sit around and think about my pain, you know. . . I don't know about how it would help somebody else, but it helps me to take my mind off my pain.

Similarly, Participant F described prayer as an approach to manage his pain: "I pray a lot. . . Gets your mind off what the hell you doing. . . I'm just a religious individual. I love God and God loves me, and he keeps me moving, honey."

The diversity of nonpharmacological approaches (from physical to passive therapies) illustrates the wide range of coping mechanisms participants experiment with to manage their pain.

**Barriers.**   Of the participants who reported using nonpharmacological therapies, nearly all reported encountering barriers including issues related to accessibility and availability. For instance, Participant C, a black man in his late-40s, benefited from physical therapy, but ultimately lost access to the service because of payer coverage limitations:

(R: While I was on codeine we did physical therapy, and after eight sessions we noticed that the [prescribed opioid] dosage was going backwards. So instead of six or eight [codeine pills per day], I was back to three. . .you know, it's lower. . . [But] right after that, the sessions stopped. They only gave me eight [sessions]. . .

I: And why did they only give you eight physical therapy sessions?

R: They said Medi-Cal only covers eight sessions.

Similar to Participant C's experience with physical therapy, Participant D, a white man in his late-30s, described benefiting from acupuncture but ultimately discontinuing the therapy because of the out-of-pocket expenses:

R: It [acupuncture] really fucking helped. They wanted me to pay money. . .There was this sliding scale thing, but I can't afford even what they were asking. Even as beneficial as I felt it was, and I don't care if it's placebo effect or what, but in my world, if I'm not hurting, and I'm taking a least amount of opiates, then I'm doing something right.

I: Have you talked to your doctor about being able to get the acupuncture prescribed?

R: Medi-Cal will not pay for that.

In both examples, participants reported that nonpharmacological therapies such as physical therapy and acupuncture decreased both pain and OPR consumption. These pain management modalities, however, were unsustainable due to administrative barriers related to payer coverage.

Participants also noted logistical barriers to accessing nonpharmacological therapies such as availability and accessibility. Participant E, a white woman in her early-50s, explained that, while acupuncture had been beneficial, its availability was limited:

[Acupuncture] was helping in the beginning. . .but. . .I can't have acupuncture any time I want it. . .I can't call somebody up and go, 'Well, I'm really in pain, it's two o'clock in the morning, can you come over and do this for me?' . . .So it works, but it doesn't work all the time.

Similarly, Participant F reported the geographic distance to pain management services as a barrier:

I met with these pain management people. . .about how to basically control it and the steps. "But we're gonna do that for you and you have to come here." It's like, "You know, you lost your mind. [Laugh.] I'm not going through all of that. Are you crazy? . . .It's not like going on. . .a freaking bus ride across the city to go to [location of pain management services] every so many hours.

## Overview of illicit substance use

The majority of participants reported using illicit substances to decrease physical pain, while some reported also using to decrease psychosocial pain and increase function. As such, many participants described their substance use as an emotionally charged experience. Participant H, a white man in his late-40s, explained this phenomenon: "I'm not looking at it recreationally. . .I just don't wanna suffer."

Other participants also described a similar feeling of desperation. Participant J, a black woman in her early-60s, describes this sensation by saying she would do, "anything [to] stop the pain, I don't care, you know."

In fact, a significant portion of participants indicated that their ideal pain management regimen was reverting back to their prior prescription. Participant K, a black woman in her mid-60s, stated:

I: What's your ideal [pain management regimen]?

R: Where my medication was a year and a half ago. . .That's when I was really ideal. . .I mean, the difference in the pain that I experience now. . .even my body is broke down.

These sentiments illustrate that, for many participants, self-managing their pain illicitly was not a preferred solution but rather one of last resort.

## Illicit OPRs

**Rationale for use.** Most participants reported their rationale for using illicit OPRs was to replicate their prior opioid prescription after having been reduced or discontinued. For instance, Participant L, a black woman in her late-50s, explained that after being reduced from

methadone prescribed for pain, she "would use the same that. . .the same thing [to] what I use [d]. I would use [the same] pills".

Another participant (Participant M–a white man in his mid-50s) described trying to replicate his prior methadone prescription after it was reduced:

> R: I'm used to the feeling that [methadone pills] gives me when I take them. . .But any other pills, no, I can't do it. . .I don't like the way it makes me feel . . .I ran out [of my prescription] a couple times. I went to the street and asked certain people and they would give them to me . . .Sometimes [I buy] three at a time, so this way I'll be able to have them. . .And never other than that [methadone pills]. Other than that, none.
>
> I: And when you were getting eight pills a day from your doctor [previous dose], at that point, were you buying pills from the street as well?
>
> R: No I wasn't . . . [I started buying] ever since he's [my provider] started lowering them [prescribed methadone], lowering, lowering them.

Here, Participant M described experimenting with other pills to identify his ideal pain management regimen, but reported ultimately seeking out methadone, the same medication he was prescribed by his provider. Participant N, a mixed-raced woman in her early-50s, also reported seeking an approximation for her prescription from the street to avoid withdrawal symptoms when she was unable to access timely refills:

> She [her provider] gives me my pain killers according to her schedule. And if you miss an appointment, they don't give you your pain killers, and they don't care if you [go into] withdrawal. I had to buy oxycodone on the street sometimes at a dollar a [milli]gram.

**Facilitators/Benefits.**   Some participants benefited from their knowledge about street-drug use and were able to manage the transition with greater ease than others. Participant D, for instance, reported understanding how to navigate the illicit market:

> I'm a gay man in San Francisco and I have an internet connection. So there's websites and apps that within five minutes of logging on you can have relations, you can have sex, and you can have whatever drug you want; five, ten minutes.

Similarly, Participant J expressed easily accessing illicit OPRs when she needed to supplement her prescription after reduction:

> I have to go buy some down in the [neighborhood], wholesale. They charge like $2, $3 a pill. . .Oh, I don't have a problem. I'm a diva! I go in all neighborhoods, I don't care how rough it is, when I go through they open up.

Participant H explained the benefit of having a "safety net" supply of illicit OPRs in case he encountered delays or gaps in his prescription:

> I: What have you been doing for the last month [during an opioid prescription gap]?
>
> R: Buying them from my friend. . .I buy two pills a week and that's eight doses, and I take one a day. I'm very good at having a back-up or the net under the wire. . .So if I fall off the wire I'll hit the net. I plan ahead. I don't wanna suffer ever.

In having a "back-up" plan, Participant H ensured that he was able to maintain his pain management regimen even amidst prescription breaks.

**Barriers.**   Accessing illicit OPRs was complicated and participants reported several barriers to navigating the underground system, including logistical (unpredictable purity, cost, and insufficient options), and knowledge-based (lack of experience and lack of risk reduction education).

Participant K described managing logistical barriers by transitioning from illicit OPRs to heroin due to fear of impurities and unreliability in the illicit OPR supply. For Participant K, a lack of trust in illicit OPRs was a motivating factor to transitioning to heroin:

> R: They started making [fake prescription] pills. . .And I won't go for that. . .They got more loyalty in the illicit than they do in the licit. I mean, in the real prescribed medication, they got more cheating going on in that.

Other participants noted cost as a barrier to obtaining illicit OPRs. For instance, Participant L bought methadone from the street after her prescription was reduced but cost ultimately became a barrier: "There's still pain, but I just can't afford to go out and pay ten dollars for a pill. I can't afford that." Instead, Participant L reported replacing methadone with alternative modalities that were more affordable including writing, coloring, reading, prayer and using crack cocaine.

Participants who lacked experience using illicit substances identified knowledge-based barriers when using illicit OPRs to manage pain. For instance, Participant H described relying on peers to access his supply because he was unfamiliar with the illicit economy:

> I got [methadone] from my friend who would go get it for me; 'cause I don't know how to buy drugs. I know how to pay for them but I don't know how to walk on the street and say, 'You, you . . .' You know, like 'Who has the crack? Who has the speed?' You know what I mean? Like I don't know how to do that.

Similarly, Participant N reported never using illicit OPRs prior to being prescribed opioids and relied on others to procure illicit OPRs when she would run out of her prescription early:

> I: You said that three months into being at [clinic] was the first time you bought pills [oxycodone] from the street. Can you tell me how you knew where to go, how you knew what to buy, what that experience was like?

> R: Actually I didn't. My home care provider [informal caregiver] saw me and he went and got them for me.

> I: Okay. Can you tell me more about that?

> R: I was screaming. I was going up on my third day and he started crying too, "You can't stay this way, when are the pills coming?" I told him probably Tuesday and this was on Sunday night and he said, "You can't stay this way." And so he went and got them.

In addition to being unable to access illicit OPRs herself because of lack of experience, Participant N also reported a substantial financial burden of purchasing oxycodone from the street: "Since I've been at [clinic], I've probably spent $7,000. . .Since I've been at [clinic] I haven't seen my kids. You know, it's all about the oxycodone. I can't save money to go visit my kids when I'm buying street drugs."

When Participant N had insufficient funds to access illicit oxycodone, she reported going through withdrawal which, in one instance, led to an overdose event:

I: Had you been using any oxycodone the days prior [to the overdose]?

R: No, there was no money.

R: I'm like going, "I can handle this." And I was taking Excedrin. . .and I told myself I had the flu. . .So I was just vomiting. . .And then finally my medication came in and like I told you I took it and I dozed off. So when I woke up I took another. And that was it. About a few minutes later I thought I was going to vomit and then boom, everything started dimming out for me. And that's weird, I've never done that before. . .It's like the lights started going out slowly, it started turning black.

### Heroin

**Rationale for use.**  While many participants described a desire to replicate their prior opioid prescription, some reported managing their pain with heroin instead. For some participants this was because they had a history of heroin use, while for others, it was a response to the barriers they encountered when attempting to access illicit opioid pills.

Participant K had a prior history of heroin use and described increasing her use to ensure her pain relief was sustained after her opioid prescription was reduced:

R: I kind of like substitute my medicine [with heroin] so it lasts. . .I make it up by doing the right dose for three days, and for two days I'll substitute the other missing portion with the heroin and it brings it up to that same level.

I: So before your opiates were reduced, how often would you say, just roughly, were you using heroin?

R: Maybe once every two or three months.

I: And then once your opiates were reduced, how often would you say you were using heroin?

R: Once every two or three days.

While some participants increased their current heroin use, others initiated heroin for the first time. Participant E, for instance, had a history of stimulant use, yet reported never having used heroin prior to when her prescription opioids were reduced, and described the transition as a last resort:

I mean, I've done pills; I've taken some because they were the thing that was there. 'Cause basically, I'm doing it for pain management. It wasn't like it, "I prefer heroin over this." But heroin's just easy to get, and not as expensive as everything else.

In addition to the ease and cost benefits, Participant E began using heroin because she was "chasing the pain relief":

Heroin was really not the appealing thing. The appealing thing was when. . .someone said, "Well, this will help your hands. This will help your knees." And. . .I did it, and it did. You know, I was nauseous and stuff but the pain was relieved really well. And unfortunately, the pain relief was almost like getting high would be to somebody else. That's what got me wanting to use it more, more, more because it wasn't getting high from it. It was because I wasn't hurting anymore . . .So that's what made me wanna do it, even though I knew that this is not a good road to go down. . .It was chasing the pain relief, not the high.

Similarly, Participant O, a white man in his mid-40s, expressed his rationale for using was because of a desire to be pain-free. Here he described watching someone fall into a heavy nod after injecting heroin:

R: It just looked like they were like really, really, excuse my language, fucked up, you know what I mean? And it looked very, very, very comfortable . . .You know what I mean? And very pain-free. . .And I wanted that, you know.

I: When you say people were. . .pain-free, what do you mean by that?

R: They were comfortably numb. . .Not numb and the feelings of like, emotionally, which I'm sure they were too, man, but. . .I just wanted not to, you know, feel pain. I mean, I've been living with pain all my life, man.

**Facilitators/Benefits.** Some participants' experiences were facilitated by their comfort accessing heroin, and their ability to independently manage and monitor their use, as Participant K described:

I knew what to do, and I knew where to go. . .And I ingested [heroin] just like I ingested with my medication to arrive at the comfort zone on legal medicine. I had to learn that same approach with my illicit drug usage.

Here Participant K illustrated that even knowledgeable users may undergo a process to learn their ideal pain management regimen. Participant K explained employing risk reduction techniques, like using a consistent and trustworthy supplier, to ensure her safety:

They [her provider] started dropping me because of the state regulatory crap. But the state don't regulate my body and it don't regulate my pain, so I regulate it. . .I do what I have to. . . .I know better than to hurt me. I know when I've reached the level and I have sense enough to find the right individual to purchase my illicit products from. . .They have the same thing, and when they change [my supply], they let me know honestly. We have that kind of rapport. It's like [my dealer is] my pharmacist and we keep it like that.

Participant K described a shadow-medical system in which she is her own advocate and pain specialist. By referring to her dealer as her pharmacist, she further emphasizes the medicalization of her heroin use.

Another reported benefit of heroin was improvement to participants' function/productivity that had declined due to unmanaged pain. Participant E described:

R: And then at about forty milligrams [of prescribed methadone, reduced from previous dose] a day I couldn't take it anymore. I couldn't do it anymore . . .

Things. . .weren't getting done because I couldn't do them. I couldn't get on my knees to. . .wash the kitchen floor well.

I: And is that because the pain was not being managed?

R: Correct. . .So then I started using so that I could get things done. So I could get to the grocery store. . .Things like that.

I: So when you say you started using then, what were you using?

R: Heroin.

Heroin was able to suppress Participant E's pain to allow her to accomplish activities of daily life like cleaning and grocery shopping.

**Barriers.** Participants described several barriers to heroin use including lack of experience, negative health effects, inaccessibility, and social stigma.

Participant P, a mixed-race man in his mid-50s, transitioned from methamphetamines to heroin and described his lack of education as a barrier to safe use:

> About four months ago I almost OD'ed . . .I decided to change my drug of choice, and I didn't know the right amount to do. I did too much. . . They had to give me a shot of Narcan [to reverse the overdose].

Participant D had a history of injection stimulant use but had never used heroin prior to his prescription reduction and described his lack of knowledge around heroin injection, including overdose risk reduction techniques such as naloxone utilization, as a barrier:

> I couldn't find any pills anywhere to make up for what she [his provider] had taken away. All of a sudden, I mean, I was sick. I'd. . . never shot up. I didn't know how to cook it [heroin] up or nothing. And believe it or not, the person that I went to ask about it, they didn't want to do it, but they were behind in rent and I had cash, and I said, "I'll pay your rent if you'll teach me how to do this. I just want to know how to do it right and not hurt myself.". . .He said, "The things that you look out for is when you have trouble breathing, Narcan yourself. If you are getting too sleepy too quickly, Narcan yourself. If you feel odd after you do a shot, Narcan yourself. If you still feel odd, and another dose does nothing, call 911." He was like, "You've been doing drugs for many years. There's no reason that you have to die because a doctor's not taking care of you.

While Participant D had access to risk reduction and overdose education, he still identified barriers around his heroin use, including social stigma:

> At the time I started having to use heroin, I had not used a syringe in almost a year. And I'd been shooting crystal meth since I was thirteen. . .It's sad that she [provider] has decided to mess with my pain medication, where [previously] I could just take a pill and things were fine, to [now] sticking myself. Heroin in the gay community [is] looked down on by other users. And no matter what your background, your education or how your life is going, you could be perfect, but they see it [heroin] is not clear like meth [and] immediately, you're not trusted anymore. There's this very bad stigma that goes along with it.

## Stimulants

**Rationale for use.** Many participants identified pain management as their primary rationale for using stimulants, as Participant M explained:

> I: When was the last time that you used cocaine?

> R: Maybe last week. . .Because I had no opiates to help the pain. No methadone. So I sniffed it [cocaine]. . .the pain went away.

Similarly, Participant P reported using methamphetamines recreationally however also noted that he increased his use when feeling pain:

I: Does it [methamphetamines] have any effect on your pain in any way?

R: It gets rid of the pain.

I: Did you change. . .the amount of meth that you used once you started feeling more pain in your hands and feet?

R: I'd do a lot. [A] whole syringe full.

Participant O also expressed feeling pain relief after using methamphetamines:

I: So you use it [methamphetamines] sometimes but not consistently.

R: Yeah. And I only use it when like . . . I miss going to the [methadone] clinic that day to actually dose, right?

I: Uh-huh.

R: For me. . .a shot of methamphetamine will sometimes be better than . . .like a pain pill to take the pain away.

I: . . .So sometimes you'll use meth when you've missed a dose and you need to control your pain?

R: And I'm starting to feel sick. And the pain's coming . . . Yeah, yeah, I'll do it like that, and it'll work better than if I took, you know, some type of pain medication to substitute; and not all doctors understand that.

Most participants who reported using stimulants had a history of stimulant use and did not seek it out explicitly to manage their pain. Nevertheless, many still identified it as a tool they used to manage their pain.

**Facilitators/Benefits.** While participants' narratives around stimulant use for pain management varied, a number of participants identified alternative benefits to their stimulant use, including managing psychosocial challenges and increasing productivity.

For instance, Participant L reported that crack was a facilitator to cope with the emotional burden of his life, connecting his use to difficulties around managing loss:

Every month it's a struggle, 'cause somebody's. . . I'll have my whole family is deceased, so each month it's a struggle; it's somebody's death anniversary. It's my birthday and everybody gone, so that's harder alone. And that's hard itself, you know.

Participant L continued to describe also using crack to increase productivity: "When I buy a piece of crack, I'm buying false energy. . .I'm trying to get something done, I use it. Just to be buying it for the hell of it? No, I buy it for a reason."

Participants identified productivity and psychosocial management as non-physical benefits of using stimulants.

**Barriers.** Counter to the descriptions above, some participants reported that stimulants were inadequate at managing pain, as described by Participant E:

Speed did some good, it helped with my pain. Back in the day, it would help with the pain, but then it would make more pain when I'd come down.

Similarly, Participant L explained that crack cocaine increased his pain:

R: If I smoke too much crack, then it's [my current opioid prescription] not enough, no.

I: So what happens when you smoke crack?

R: When you smoke too much crack? It take the methadone out of your system and make the pain, you know, makes it more painful.

While many participants used stimulants to manage pain, its success varied, and for many, feeling increased pain as stimulants wore off was a significant barrier.

## Discussion

After being reduced/discontinued from prescribed OPRs, many participants reported being left out of the traditional medical system and tasked with becoming their own pain management specialists: independently assessing their pain and developing informal pain management solutions including nonpharmacological therapies and illicit OPRs, heroin and stimulant use. Most participants described embarking on thoughtful and intentional journeys of self-managing pain that included multimodal experimentation. Cost, payer coverage and the geo-location of nonpharmacological interventions served as structural barriers that hindered initiation and ongoing access to nonpharmacological treatments. Most participants who turned to illicit substances did so after exhausting other options.

OPR reductions/discontinuations may be particularly traumatizing and physically taxing for patients who have been maintained on opioids for long periods of time, due to significant and difficult-to-reverse changes in body function, neuroplasticity, and the physical and psychological perception of pain [5,30]. Among many barriers to safely tapering opioid-experienced patients is protracted abstinence syndrome, which is caused by allostatic changes due to opioid tolerance and/or dependence, and can result in extended withdrawal symptoms (e.g. anxiety, depression, fatigue, increased pain etc.) that last for years and may have substantial negative consequences on patients' physical, psychological, and psychiatric wellbeing [3,5,31,32]. Although our participants did not directly link their pain or pain management to HIV, they may be more likely than other patients to have a long history of receiving opioid therapy and thus face more challenges with transitioning off of opioids [23,33]. The therapeutic and clinical complexities that make opioid-experienced patients distinct from their opioid-naïve counterparts should be accounted for in the development of opioid stewardship guidelines [3,5,14]. The FDA has recognized the dangerous and potentially life-threatening consequences of uniform tapering guidelines or abrupt tapers for opioid-experienced patients and now recommends gradual, individualized tapering plans built in conjunction with patients to modulate risk of uncontrolled pain, psychological distress, withdrawal symptoms, emergence or reemergence of substance use disorder, violence, or suicide [3,30].

Patient's experiences of pain and capacity to access therapies are multifaceted and complex. Recognizing the various issues that may affect how a patient is able to manage physical and psychological pain may help providers manage long-term opioid therapy. Considering opioid experience, substance use, and social and psychosocial supports are essential prior to discussing a taper of prescribed OPRs. Providers should recognize the significant impact that any reduction in OPR prescriptions may have on a patients, as some may begin supplementing prescriptions with alternative approaches including street drugs during prescription gaps or tapers, which complicates pain management efforts and clinical evaluation [3,5,30]. To manage opioid-experienced patients, it is critical for providers to develop patient-centered pain management approaches and, if indicated, opioid reduction plans, in conjunction with patients, and to pay close attention to patients' well-being during this vulnerable period.

## Limitations

Our study has several limitations. First, these data were collected via self-report during in-person interviews, which may lead to social-desirability or recall biases. Second, our analysis was driven by a small sample size which did not include provider perspectives. Third, participants were HIV-positive and had a history of some substance use. While patients did not associate their HIV with their pain management or their opioid reduction/discontinuation their experiences still may not be generalizable to a wider audience, including those patients who do not have a history or substance and those living without HIV. While participants did not associate HIV with pain or pain management, they may be unaware of certain factors that influence opioid effects such as the duration of treatment with opioid pain relievers. Finally, we did not include themes related to the use of punitive tools around opioid prescribing which, while an important theme emerging in today's literature, was not the focus of our analysis.

## Conclusions

When losing access to opioid therapy for chronic non-cancer pain, HIV-positive patients with a history of substance use pro-actively self-manage symptoms outside of the traditional medical system. In doing so, participants describe a range of rationales, facilitators and barriers that they experienced while transitioning to non-pharmacological therapies, illicit OPRs, heroin and/or stimulants. Patients may experiment with a range of pain management modalities including nonpharmacological therapies and illicit substance use. For this patient population, illicit drugs are a common remedy, including both opioids and stimulants. When providers make any change to patients' long-term opioid therapy, a holistic and patient-centered approach should be considered.

## Author Contributions

**Conceptualization:** Emily Behar, Kelly Knight, Phillip O. Coffin.

**Data curation:** Emily Behar, Rita Bagnulo.

**Formal analysis:** Emily Behar, Rita Bagnulo.

**Funding acquisition:** Phillip O. Coffin.

**Methodology:** Emily Behar.

**Project administration:** Emily Behar.

**Resources:** Emily Behar.

**Supervision:** Emily Behar, Phillip O. Coffin.

**Writing – original draft:** Emily Behar, Rita Bagnulo, Phillip O. Coffin.

**Writing – review & editing:** Emily Behar, Rita Bagnulo, Kelly Knight, Glenn-Milo Santos, Phillip O. Coffin.

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
