## [Decision Letter · Decision Letter 0]

22 Aug 2019

PONE-D-19-16726

“Chasing the pain relief, not the high”: experiences managing pain after opioid reductions among patients with HIV and a history of substance use

PLOS ONE

Dear Ms. Behar,

Thank you for submitting your manuscript to PLOS ONE. After careful consideration, we feel that it has merit but does not fully meet PLOS ONE’s publication criteria as it currently stands. Therefore, we invite you to submit a revised version of the manuscript that addresses the points raised during the review process.

This manuscript promises to be a timely contribution on a topic of high relevance. It received two generally positive reviews, with insights on how the manuscript can be approved in terms of substance and structure. We would appreciate receiving your revised manuscript by Oct 06 2019 11:59PM. To enhance the reproducibility of your results, we recommend that if applicable you deposit your laboratory protocols in protocols.io, where a protocol can be assigned its own identifier (DOI) such that it can be cited independently in the future. For instructions see: http://journals.plos.org/plosone/s/submission-guidelines#loc-laboratory-protocols

We look forward to receiving your revised manuscript.

Kind regards,

Leo Beletsky

Academic Editor

PLOS ONE

3. Please amend your manuscript to include your abstract after the title page.

Reviewers' comments:

Reviewer's Responses to Questions

**Comments to the Author**

1. Is the manuscript technically sound, and do the data support the conclusions?

Reviewer #1: Yes

Reviewer #2: Yes

2. Has the statistical analysis been performed appropriately and rigorously? 

Reviewer #1: Yes

Reviewer #2: N/A

3. Have the authors made all data underlying the findings in their manuscript fully available?

Reviewer #1: Yes

Reviewer #2: No

4. Is the manuscript presented in an intelligible fashion and written in standard English?

Reviewer #1: Yes

Reviewer #2: Yes

5. Review Comments to the Author

Reviewer #1: This is a qualitative analysis of interviews with individuals with HIV who experienced after long term opioid pain medication reduction or discontinuation. This is an important contribution to the field and is generally a well-done analysis with an accompanying well-written manuscript . A few comments for the authors to consider:

1)

Abstract:

Consider reporting Ns rather than percentages in the results paragraph (see related comment regarding Results below).

Intro:

-The final paragraph in the introduction (line 49) would benefit from introducing the participant population as having a history of illicit substance use in addition to HIV and chronic pain, as this is an important differentiating factor for the sample.

Methods:

-Please provide a citation for the longitudinal COPING study cohort, if one exists.

-Please provide additional information about how OPR dose reduction or discontinuation was defined? Participant self report vs chart review?

-Please provide some additional information about how/when interim analyses were conducted, which would allow the investigators to know when thematic saturation was reached.

-Please provide citations for content analysis methods (line 79).

Results:

-Given that this is a qualitative analysis with an appropriately small N, I would recommend reporting the Ns for each demographic characteristic (ie 11 were male) rather than reporting percentages, which may suggest to readers that these findings are more representative/generalizable to a wider population, which we are not sure they are.

-Are you able to provide information regarding how many participants experienced dose reductions vs those who experienced discontinuation? Are you able to provide any info regarding average size of the dose reduction or discontinuation?

-Consider expanding discussion of the fact that HIV-related factors are not included in the qualitative analysis (line 95). Was it necessary to enroll HIV infected participants or do you believe that the results would have been similar with a non-HIV infected population. Please provide further justification one way or the other.

-Please remove marijuana from the non-pharmacologic therapies section, as cannabis and its derivatives are increasingly used in medical/pharmacologic contexts. Consider referring to the substance as cannabis, which may have fewer stigmatizing connotations relative to marijuana.

-Some participants quotes (participant F, for example, in the non-pharmacologic section) seem over-represented. If possible, consider diversifying the participants whose quotes are provided in the manuscript to provide a more comprehensive overview of perspectives.

Conclusions:

-I would encourage the authors do use the conclusion to further contextualize their work given how unique it is for them to be providing detailed perspectives from individuals with substance use/HIV. These are individuals whose voices are not commonly heard in the research literature, making this a particularly important contribution.

-Seems worth revisiting here the fact that the common tropes were unrelated to HIV infection or HIV-specific care. Can you please provide some additional context to the brief discussion of this in the results section (line 95), see comment above.

-Can the authors please provide some further clarification and summarization about the respective rationales, facilitators, and barriers to the pharmacologic, non-pharmacologies and illicit substance use self-management strategies participants reported?

Reviewer #2: Overall this was an excellent paper and interesting to review. I learned a lot about the patient experience following opioid taper or cessiation that will impact my clinical practice. I do have some suggestions prior to publication for the authors. One thing that struck me was that there was no mention of fentanyl in the participant quotes. In other parts of North America the risk of fatal overdose is so great as the heroin supply has been replaced by fentanyl. This makes the results of your study even more compelling. I am not sure how to fit this in but it is an observation I wanted to share.

General comments:

Is the period supposed to be before or after the reference parenthesis? I would check with the journal, usually I have seen the period before.

Abstract:

In the background or methods please state the aim of the study.

Background

Line 21-24: I think justified is a strong word. Additionally, there is also epidemiological evidence that showed as the rate of opioid prescriptions increase so too did the rate of opioid misuse. This also supported this public health strategy

Line 24: suggest rephrasing “However, reduction in prescribing also carry risks:” to “However, decreasing the opioid dose or prescription cessation also carries risks:”

Line 27: why is it paradoxical?

Line 43-47: The first sentence is slightly repetitive from the sentence in line 24 and could be removed or included above in 24. It is reasonable to end paragraph starting on line 30, with the second sentence of this small paragraph.

Overall comment for the background: Has there been any other qualitative research in this area? If so how does your work build or add to this? If not, it is reasonable to say this, and would strongly support the need for this work. To accommodate these extra sentences you could consider paring down the second paragraph.

Methods

Data collection and analysis

Line 62: Could you say what the COPING study was about or at least what the acronym stands for and a reference to the original publication or study protocol?

Line 71: Please describe how the interview guides were developed. Was it a flexible guide etc?

General comment: How was language addressed? Were non-English speakers included?

Results

The table and quantitative summary were clear and helpful. Overall the results were presented very clearly and the quotes chosen highlight the themes very well.

Line 281: Suggest starting the sentence at Participants.

Line 288-90: This quote is a bit confusing and does not strongly support the above theme. It almost sounds like he trusts the illicit market more? I re-read it several times. Perhaps it would be clearer if more of the quote was included or the interviewer question? Alternatively, perhaps there is another quote that could be used.

Line 329-336: Does the rest of the quote describe her overdose experience and how she survived? If so would be interesting to include this if it does.

Line 402-407: This quote does not clearly capture the concept of harm reduction techniques of tester shots or cautious dosing. I suggest removing this idea or including more of this quote or another quote that illustrated this concept more clearly.

Line 426: suggest replacing “household chores” with “activities of daily living”.

Line 434: I would rephrase the idea of “education” to “lack of experience with heroin” as this seems to be what the quote supports.

Line 447-452: This part of the quote is very interesting but does not connect to the statement above. I wonder if there is a way to highlight this piece of Narcan training specifically?

Discussion

551-557: This seems like a new theme not previously discussed or supported by the quotes in the paper. Participants did not describe withdrawal as a driver for their using illicit opioids. I suggest this be removed or add some quotes that support this concepts.

6. PLOS authors have the option to publish the peer review history of their article (what does this mean?). If published, this will include your full peer review and any attached files.

Reviewer #1: Yes: Benjamin Bearnot, MD, MPH, FASAM

Reviewer #2: No

---

## [Author Response · Author response to Decision Letter 0]

3 Oct 2019

Dear editors and reviewers,

The authors would like to thank you for the thoughtful review of our manuscript. Because of your comments, we feel that the manuscript is clearer and more sophisticated, particularly in terms of the discussion around HIV. We have attached our comments in the Response to Reviewer file in the attachments. We hope you will find our feedback and revisions acceptable. 

Thank you,

Emily Behar, on behalf of the entire author team

---

## [Editor Report · Decision Letter 1]

12 Nov 2019

PONE-D-19-16726R1

“Chasing the pain relief, not the high”: experiences managing pain after opioid reductions among patients with HIV and a history of substance use

PLOS ONE

Dear Ms. Behar,

Thank you for submitting your manuscript to PLOS ONE. After careful consideration, we feel that it has merit but does not fully meet PLOS ONE’s publication criteria as it currently stands. Therefore, we invite you to submit a revised version of the manuscript that addresses the points raised during the review process.

We would appreciate receiving your revised manuscript by Dec 27 2019 11:59PM. To enhance the reproducibility of your results, we recommend that if applicable you deposit your laboratory protocols in protocols.io, where a protocol can be assigned its own identifier (DOI) such that it can be cited independently in the future. For instructions see: http://journals.plos.org/plosone/s/submission-guidelines#loc-laboratory-protocols

We look forward to receiving your revised manuscript.

Kind regards,

Leo Beletsky

Academic Editor

PLOS ONE

Additional Editor Comments (if provided):

This piece is much improved after the revision. Acceptance is contingent on the following minor revisions:

A. Overall comment/suggestion: may be useful to get a sense of the general prevalence of perspectives in each category, ie how pervasive the view listed was, and whether there were contrasting views and experiences. Obviously no need to quantify, but something more descriptive than the current "many" could help orient the reader.

B. Overall comment/suggestion: may be enriching to foreground criminalization and punitive measures in the health care system, either in the narratives listed, in the discussion, or both. Feels a bit like the elephant in the room that the manuscript glosses over.

Specific comments:

- Line 37: although certainly useful in some cases, multimodal pain therapies described do not, as a rule, have a more robust evidence base when compared with opioid pharmacotherapy

- Lines 42-43: clarify that transition may, in fact, be facilitated by tapering interventions or other measures to suppress pharmacotherapy access (See, e.g. Larochelle et al, 2019)

- Lines 51-53: what is the prevalence of injection drug use in this population? Injection contributes additional vulnerability to chronic and acute pain

- Line 66: how was Rx history determined in the larger study?

- Line 81: double check Journal formatting rules for citation of software packages
---

## [Author Response · Author response to Decision Letter 1]

3 Dec 2019

PONE-D-19-16726R1

“Chasing the pain relief, not the high”: experiences managing pain after opioid reductions among patients with HIV and a history of substance use

PLOS ONE

Editor Comments

This piece is much improved after the revision. Acceptance is contingent on the following minor revisions:

A. Overall comment/suggestion: may be useful to get a sense of the general prevalence of perspectives in each category, ie how pervasive the view listed was, and whether there were contrasting views and experiences. Obviously no need to quantify, but something more descriptive than the current "many" could help orient the reader.

In an effort to not quantify the data, the authors were unable to demonstrate prevalence of perspectives, as the editor anticipated. We did, however, find some themes more emergent than others and tried to reflect that as best we could in the manuscript. This is, of course, an age-old challenge in qualitative research. In order to make our findings more “quantifiable” we changed language whenever possible to give a better sense of consistency of emergent themes. The changes are listed below.

• Pg 5, line 97, included the word “vast” to demonstrate the overwhelming consistency of agreement.

• Pg 5, line 114, included the word “most” to emphasize general consistency of reported nonpharmacological therapies

• Pg 6, lines 138-139 now reads: “The majority of participants who used nonpharmacological therapies found that they reduced both pain and opioid intake.”

• Pg 7, lines 164-165 now reads: “Of the participants who reported using nonpharmacological therapies, nearly all reported encountering barriers including issues related to accessibility and availability.”

• Pg 9, line 217 now reads, “Other participants also described…”

• Pg 9, line 221 now reads, “In fact, a significant portion of participants indicated that their ideal pain management…”

• Pg 10, line 234, change the word “many” to “most”

B. Overall comment/suggestion: may be enriching to foreground criminalization and punitive measures in the health care system, either in the narratives listed, in the discussion, or both. Feels a bit like the elephant in the room that the manuscript glosses over.

• We agree that this is an important issue, and one that came up in some of our interviews. Unfortunately, it would not be appropriate to include such a significant theme in this paper given that the focus of our analysis (and interviews) was on patients’ pain management experiences after reduction/discontinuation so did not go into extensive detail about punitive measures during their opioid prescribing period. There is, however, a substantial body of literature emerging on this issue and we feel that other researchers have covered this in more depth than we could do here. We thought it most fitting to include a description of this in our limitations section, which now reads: “Finally, we did not include themes related to the use of punitive tools around opioid prescribing which, while an important theme emerging in today’s literature, was not the focus of our analysis.”

Specific comments:

Line 37: although certainly useful in some cases, multimodal pain therapies described do not, as a rule, have a more robust evidence base when compared with opioid pharmacotherapy

• The authors agree and want to make sure that point is clear in the paper. Pg 2, lines 33-34 now reads: “While evidence about the effectiveness of these modalities is limited compared to opioid pharmacotherapy, a systematic review of noninvasive nonpharmacological treatments for chronic pain suggests that exercise, multidisciplinary rehabilitation, acupuncture, cognitive behavioral therapy, and mind-body practices were associated with improvements in pain and function among patients with selected chronic pain conditions.”

Lines 42-43: clarify that transition may, in fact, be facilitated by tapering interventions or other measures to suppress pharmacotherapy access (See, e.g. Larochelle et al, 2019)

• Pg 3, lines 44-45: “Retrospective research has found that patients with a history of substance use (15–17), on a high daily dose of opioids (2), and with multiple pain complaints may be at heightened risk of transitioning to illicit opioids (18); some of these transitions may be iatrogenically facilitated by tapering interventions or other measures to suppress access to pharmacotherapy. (19)” 

Lines 51-53: what is the prevalence of injection drug use in this population? Injection contributes additional vulnerability to chronic and acute pain

• Pg 3, lines 51-52 now read: “PLWH also have unique causes of pain, such as HIV-associated neuropathy, and higher prevalence of substance use disorders and drug injection which may also contribute to higher chronic and acute pain.” 

Line 66: how was Rx history determined in the larger study?

• Pg 3-4, lines 69-70 now read “Participants’ opioid prescribing history was determined through an extensive medical record chart abstraction conducted by study staff.”

Line 81: double check Journal formatting rules for citation of software packages

• Pg 4, line 84. PLOS One instructions for reporting software state, “List the name and version of any software package used, alongside any relevant references.” We kept our citations as originally submitted as we believe our citation follows this guidance. https://journals.plos.org/plosone/s/submission-guidelines

---

## [Decision Letter · Decision Letter 2]

2 Mar 2020

“Chasing the pain relief, not the high”: experiences managing pain after opioid reductions among patients with HIV and a history of substance use

PONE-D-19-16726R2

Dear Dr. Behar,

We are pleased to inform you that your manuscript has been judged scientifically suitable for publication and will be formally accepted for publication once it complies with all outstanding technical requirements.

With kind regards,

Leo Beletsky

Academic Editor

PLOS ONE

Additional Editor Comments (optional):

Given that the first version was deemed to require only a minor review, and Reviewer 2 has favorably rated the revision, I am satisfied that the authors have adequately met concerns. I am proceeding with accepting the paper, pending additional minor tweaks as suggested by the reviewer's comments.

Reviewers' comments:

Reviewer's Responses to Questions

**Comments to the Author**

1. If the authors have adequately addressed your comments raised in a previous round of review and you feel that this manuscript is now acceptable for publication, you may indicate that here to bypass the “Comments to the Author” section, enter your conflict of interest statement in the “Confidential to Editor” section, and submit your "Accept" recommendation.

Reviewer #2: (No Response)

2. Is the manuscript technically sound, and do the data support the conclusions?

Reviewer #2: Yes

3. Has the statistical analysis been performed appropriately and rigorously? 

Reviewer #2: Yes

4. Have the authors made all data underlying the findings in their manuscript fully available?

Reviewer #2: No

5. Is the manuscript presented in an intelligible fashion and written in standard English?

Reviewer #2: Yes

6. Review Comments to the Author

Reviewer #2: The manuscript was great to read again, I have a few additional minor suggestions for your consideration, but believe it is appropriate to publish at this time

Abstract:

Background: The aim sentence is long, perhaps could end at discontinuations

Methods/results: it is unclear why/how you arrived at 18 interviews, from the paper we know this was because themaptic satuation was reached but this would be helpful to have stated in the abstract.

Methods:

Study sample: again, is there a protocol published for COPING? if so please include this reference, if not ignore this comment.

Results:

Demographics: I echo revierwers 1's comments regading specifics about the patients OPR reductions. Do you know how many of your participants had their OPR stopped, or the mean reudciton amount? From a clinical perspective, this helps providers reading your study understand how specific treatment changes impacted your study participants experiences.

7. PLOS authors have the option to publish the peer review history of their article (what does this mean?). If published, this will include your full peer review and any attached files.

Reviewer #2: No

---

## [Editor Report · Acceptance letter]

12 Mar 2020

PONE-D-19-16726R2 

“Chasing the pain relief, not the high”: experiences managing pain after opioid reductions among patients with HIV and a history of substance use 

Dear Dr. Behar:

I am pleased to inform you that your manuscript has been deemed suitable for publication in PLOS ONE. Congratulations! Your manuscript is now with our production department. 

With kind regards,

on behalf of

Prof. Leo Beletsky 

Academic Editor

PLOS ONE